# Outcomes of Endoscopic Ultrasound-Guided Biliary Drainage in a General Hospital for Patients with Endoscopic Retrograde Cholangiopancreatography-Difficult Transpapillary Biliary Drainage

**DOI:** 10.3390/jcm10184105

**Published:** 2021-09-11

**Authors:** Naosuke Kuraoka, Satoru Hashimoto, Shigeru Matsui, Shuji Terai

**Affiliations:** 1Department of Gastroenterology, Saiseikai Kawaguchi General Hospital, 5-11-5 Nishikawaguchi, Kawaguchi, Saitama 332-8558, Japan; shashim@saiseikai.gr.jp (S.H.); mshigerum@saiseikai.gr.jp (S.M.); 2Department of Gastroenterology and Hepatology, Graduate School of Medical and Dental Sciences, Niigata University, 1-757 Asahimachidori, Niigata 951-8510, Japan; terais@med.niigata-u.ac.jp

**Keywords:** EUS-BD, EUS-CDS, EUS-HGS, biliary stenosis, ERCP

## Abstract

Endoscopic ultrasound-guided biliary drainage (EUS-BD) has been developed as an alternative treatment for percutaneous transhepatic biliary drainage for patients with bile duct stenosis. At specialized hospitals, the high success rate and effectiveness of EUS-BD as primary drainage has been reported. However, the procedure is highly technical and difficult, and it has not been generally performed. In this study, we retrospectively examined the effectiveness of EUS-BD in ERCP-difficult patients with distal bile duct stenosis. We retrospectively examined 24 consecutive cases in which EUS-BD was performed at our hospital for distal bile duct stenosis from October 2018 to December 2020. EUS-guided choledochoduodenostomy (EUS-CDS) was selected for cases that could be approached from the duodenal bulb, and EUS-HGS was selected for other cases. In the EUS-CDS and EUS-HGS groups, the technical success rates were 83.3% (10/12] and 91.7% (11/12], respectively. An adverse event occurred in one case in the EUS-CDS group, which developed severe biliary peritonitis. The stent patency period was 91 and 101 days in the EUS-CDS and EUS-HGS groups, respectively. EUS-BD for ERCP-difficult patients with distal bile duct stenosis is considered to be an effective alternative for biliary drainage that can be performed not only in specialized hospitals but also in general hospitals.

## 1. Introduction

Endoscopic ultrasound-guided biliary drainage (EUS-BD) has been developed as an alternative treatment to percutaneous transhepatic biliary drainage (PTBD) for patients with difficult or unsuccessful transpapillary biliary drainage using endoscopic retrograde cholangiopancreatography (ERCP) [1,2]. EUS-BD, which is performed in many specialized hospitals, is generally classified into EUS-guided choledochoduodenostomy (EUS-CDS) and EUS-guided hepaticogastrostomy (EUS-HGS). At these hospitals, it has been reported that performing EUS-BD as the primary drainage is effective and had a high success rate. However, at present, the procedure is still highly technical and difficult, and it has not been widely used in other medical facilities [3,4,5,6]. Therefore, the success rate and safety of EUS-BD in general hospitals are currently unknown. In this study, we retrospectively examined the effectiveness of EUS-BD for patients with distal bile duct stenosis at our institution in whom performing an ERCP was difficult.

## 2. Materials and Methods

We retrospectively examined 24 consecutive cases in which EUS-BD was performed at our hospital for distal bile duct stenosis from October 2018 to December 2020. All cases underwent EUS-BD instead of ERCP due to the difficult procedure of the latter. EUS-CDS was performed for cases that could be approached from the duodenal bulb, whereas EUS-HGS was selected for other cases. The endoscopic procedures were performed by a skilled endoscopist who had more than 1000 cases of ERCP experience, more than 1000 cases of observation EUS experiences and more than 200 cases of EUS-guided fine needle aspiration.

This study was approved by the Institutional Review Board of the Saiseikai Kawaguchi General Hospital. The primary endpoint of this study was the technical success rate of EUS-BD, and the secondary endpoints were the rate of adverse events, stent patency period, and re-intervention.

For the EUS-CDS procedure, the extrahepatic bile duct was visualized from the duodenal bulb using a linear endoscopic ultrasound (GF-UCT260; Olympus Medical Japan, Tokyo, Japan). The extrahepatic bile duct was punctured using a 19-G puncture needle, and a guidewire was placed in the bile duct. The fistula was then dilated using a dilation device, and a plastic stent (PS) or a self-expandable metal stent (SEMS) was deployed (Figure 1).

For the EUS-HGS procedure, the intrahepatic bile ducts (B2 or B3) were visualized from the stomach using linear endoscopic ultrasound (GF-UCT260; Olympus Medical Ja-pan, Tokyo, Japan). The intrahepatic bile duct was punctured with a 19-G or 22-G puncture needle, and a guidewire was placed. After cholangiography, fistula dilation was performed using a dilation device, and the PS or SEMS was deployed. Alternatively, the PS or SEMS was placed without dilating the fistula (Figure 2).

Technical success was defined as a case in which PS or SEMS could be placed during EUS-BD. Clinical success was defined as a case in which cholangitis was alleviated or the total bilirubin level improved. Adverse events were defined as all complications that occurred after procedure (e.g., bleeding, stent migration, peritonitis, stent dysfunction, and so on). Early and late complications were defined as adverse events that occurred <30 days and >30 days after treatment, respectively. Re-intervention was defined as performing another endoscopic or percutaneous treatment due to recurrence of cholangitis or obstructive jaundice. The stent patency period was defined as the period before the recurrence of cholangitis and jaundice. Patients who died due to the current illness were treated as censored. The performance status refers to the Eastern Cooperative Oncology Group performance status [7]. The adverse events were graded using the American Society for Gastrointestinal Endoscopy lexicon severity grading system [8].

A specialized hospital was defined as a university hospital, a cancer center, and a tertiary medical institution. On the other hand, a general hospital was defined as a medical institution other than university hospitals, cancer centers, and tertiary medical institutions. Saiseikai Kawaguchi General Hospital was defined as a general hospital.

The results are presented as numerical values (%), while continuous variables are presented as median values (range). This study performed an intention-to-treat analysis, and the median of the stent patency periods was calculated using the Kaplan–Meier method. All statistical analyses were performed using the Statistical Product and Service Solutions (SPSS, IBM, Tokyo, Japan).

## 3. Results

Among the cases in which ERCP was performed for distal bile duct stenosis from October 2018 to December 2020, 24 consecutive ERCP-difficult cases in which EUS-BD was instead performed were retrospectively examined.

### 3.1. Patient Characteristics

The EUS-CDS and EUS-HGS groups had 12 patients each. Pancreatic cancer was the most common background disease in both groups (EUS-CDS: 9/12, 66.7%; EUS-HGS: 7/12, 58.3%). The mean total bilirubin level before treatment was 9.49 mg/dL in the EUS-CDS group and 6.72 mg/dL in the EUS-HGS group. In the EUS-CDS group, an extrahepatic bile duct was punctured, and the average bile duct diameter was 14.9 mm. The average diameter of the intrahepatic bile duct in the EUS-HGS group was 5.1 mm. In most cases, 19-G needles were used as puncture needles in both the EUS-CDS and EUS-HGS groups; 22-G needles were used in two cases in the EUS-HGS group (Table 1).

### 3.2. Outcomes

The technical success rates in the EUS-CDS and EUS-HGS groups were 10/12 (83.3%) and 11/12 (91.7%), respectively. Both groups were effective in stent-placeable cases, and the clinical success rates were similar. SEMS was deployed in 9 of the 10 cases in the EUS-CDS group in which the stent could be inserted. In the EUS-HGS group, SEMS was deployed in four cases whereas PS was deployed in the rest. Regarding fistula dilation, electrocautery dilation was performed in the EUS-CDS group, while non-electrocautery dilation was performed in the EUS-HGS group. In addition, non-dilation of the fistula was also performed in two cases in the EUS-HGS group (Table 2). An adverse event occurred in only one case in the EUS-CDS group, which developed severe biliary peritonitis (Table 3). The median stent patency period was 91 days and 101 days in the EUS-CDS and EUS-HGS groups, respectively, showing no significant difference (Figure 3). After stent insertion, resection and chemotherapy was performed in two and five patients (41.7%), respectively, in the EUS-HGS group. Meanwhile, chemotherapy was administered to five patients in the EUS-CDS group. Re-intervention was performed in five patients in the EUS-CDS group, and the technical success rate was 100%. In the EUS-CDS group, re-intervention was performed by replacing the stent or changing the position of the stent. In the EUS-HGS group, re-intervention was performed in six patients, and the technical success rate was also 100%. However, a new EUS-HGS route was required in one patient (Table 4).

## 4. Discussion

EUS-BD has been developed as an alternative treatment for ERCP-difficult cases. After Wiersema et al. reported cholangiography as an endosonography-guided cholangiopancreatography in 1999, Giovannini et al. first reported biliary drainage as EUS-guided bilioduodenal anastomosis in 2001 [9,10]. Since then, numerous EUS-BDs have been performed at specialized facilities, with some EUS-BDs being reported as the primary drainage as well as an alternative treatment for ERCP-difficult patients [3,4,5,6]. However, in many facilities, the EUS-BD has not been introduced, which hinders the widespread and generalized use of the procedure.

This study examined the initial results after the introduction of EUS-BD. Most of the target patients had malignant biliary stenosis. In addition, many of them were elderly with a decreased performance status, and there were many cases in which chemotherapy could not be initiated. The technical success rates of EUS-CDS and EUS-HGS was reported to be 90.9–100% [3,4,5,6] and over 90% [11,12,13,14]. In our study, the technical success rates for EUS-CDS and EUS-HGS were 83.3 and 91.7%, respectively, which is considered to be equivalent to previous studies. During the introduction of EUS-HGS, biliary drainage was performed by using a PS, but the placement of SEMS increased with the advent of small-diameter stents. In addition, many EUS-CDS detentions were performed using SEMS. In our study, EUS-CDS showed severe bile leakage in one case. Although the incidence of adverse events was 8.3% in EUS-CDS, only one case of adverse events was severe bile leakage. On the other hand, in EUS-HGS, no adverse events were observed.

Regarding the stent patency period, the median patency periods were 91 and 101 days in the EUS-CDS and EUS-HGS groups, respectively. The reason for the short patency period of the EUS-CDS group was considered to be due to the stent direction being changed early after the procedure. As mentioned above, the introduction of EUS-BD has been highly successful in many specialized hospitals, but the current situation is that the procedure has not been generally applied. The results at our hospital, which is a general hospital, are the initial results of the introduction of EUS-BD, which can be an index for facilities considering the introduction of EUS-BD. The technical success, which is the primary endpoint, was as high as previously reported. The rate of adverse events was low enough, and re-intervention was possible in this study. Although this study was a small number of case studies, no adverse events were observed in EUS-HGS. EUS-HGS is possibly more secure in initial introduction of EUS-BD than EUS-CDS.

In addition, EUS-BD cannot be performed by a skilled endoscopist alone, and an assistant who is familiar with both the procedure and biliary tract treatment tools and equipment is required. The advent of EUS-BD-dedicated devices may alleviate these difficulties. Recently, EUS-CDS using lumen-apposing metal stents that has a high technical success rate and low rate of adverse events has been reported [15,16,17].

This study had several limitations. This was a retrospective single-center study in which an endoscopic procedure was performed by a skilled endoscopist. Moreover, there were a small number of case studies. In the future, in order to generalize our results and the application of EUS-BD, it is necessary to carry out prospective multicenter research.

## 5. Conclusions

In conclusion, EUS-BD in ERCP-difficult patients with distal bile duct stenosis is considered to be an effective alternative for biliary drainage that might be possibly performed not only in specialized hospitals but also in general hospitals. However, adverse events have been observed, and the development of EUS-BD dedicated devices is desirable for the general application of this procedure.

## Figures and Tables

**Figure 1 jcm-10-04105-f001:**
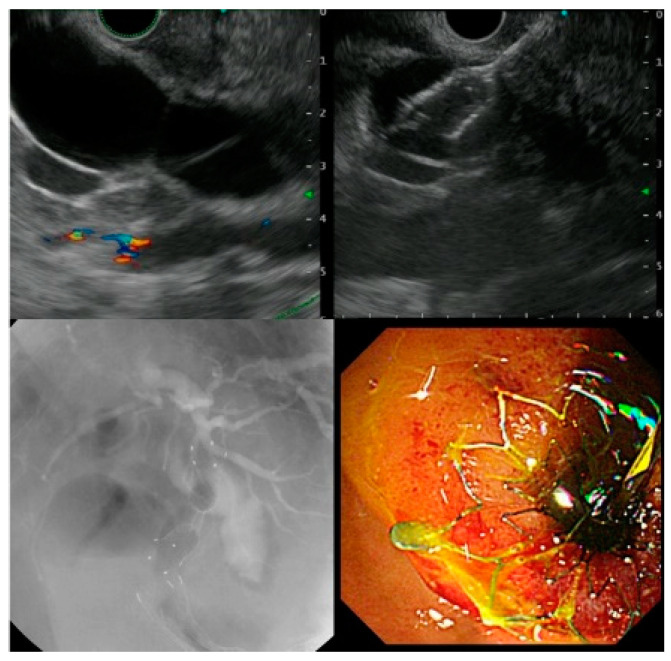
Procedure of endoscopic ultrasound-guided choledochoduodenostomy.

**Figure 2 jcm-10-04105-f002:**
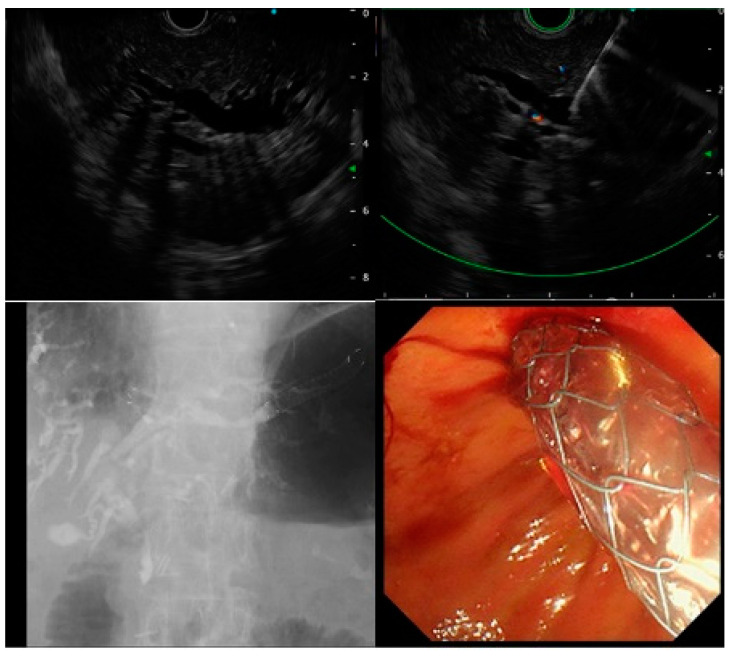
Procedure of endoscopic ultrasound-guided hepaticogastrostomy.

**Figure 3 jcm-10-04105-f003:**
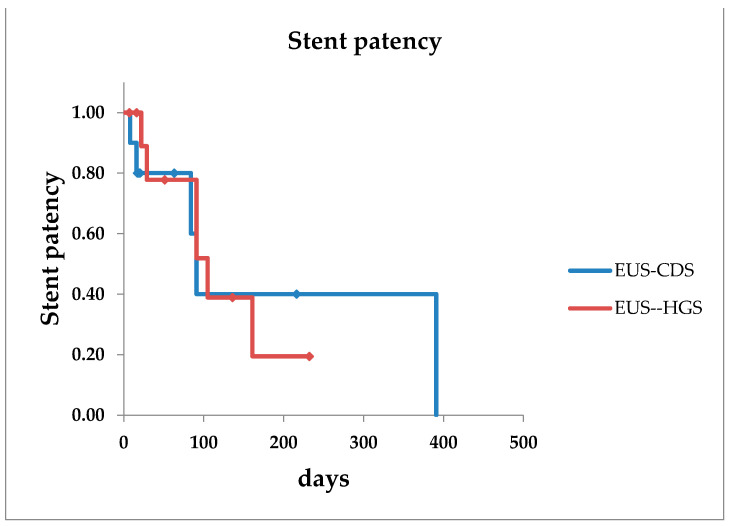
Stent patency of EUS-CDS and EUS-HGS. The stent patency period is 91 days in the EUS-CDS group and 101 days in the EUS-HGS group. EUS-CDS: endoscopic ultrasound-guided choledochoduodenostomy, EUS-HGS: endoscopic ultrasound-guided hepaticogastrostomy.

**Table 1 jcm-10-04105-t001:** Patient characteristics.

	EUS-CDS (*n* = 12)	EUS-HGS (*n* = 12)
Median age (range)	76.5 (57–75)	76.5 (60–85)
Sex, male, *n* (%)	4 (33.3)	7 (58.3)
Diagnosis, *n* (%)		
Pancreatic cancer	9 (66.7)	7 (58.3)
Cholangiocarcinoma	0	1 (8.3)
Dissemination of cancer	3 (33.3)	2 (16.7)
Bile duct stones	0	2 (16.7)
Pre-total Bilirubin, mean, mg/dL (range)	9.49 (3.29–16.06)	6.72 (0.48–13.65)
Performance status (PS)		
Performance status < 2, *n* (%)	7 (58.3)	8 (66.7)
Performance status > 3, *n* (%)	5 (41.7)	4 (33.3)
Diameter of bile duct		
Extrahepatic bile duct, mean, mm (range)	14.9 (11–20)	N/A
Intrahepatic bile duct, mean, mm (range)	N/A	5.1 (2.6–7)
Size of needle		
19G, *n* (%)	12 (100)	10 (83.3)
22G, *n* (%)	0	2 (16.7)

EUS-CDS: endoscopic ultrasound-guided choledochoduodenostomy; EUS-HGS: endoscopic ultrasound-guided hepaticogastrostomy; PS: plastic stent.

**Table 2 jcm-10-04105-t002:** Outcomes of EUS-CDS and EUS-HGS.

	EUS-CDS (*n* = 12)	EUS-HGS (*n* = 12)
Technical success, *n* (%)	10/12 (83.3)	11/12 (91.7)
Clinical success, *n* (%)	10/12 (83.3)	11/12 (91.7)
Deployment of SEMS, *n* (%)	9 (75)	4 (33.3)
Deployment of PS	1 (8.3)	7 (58.3)
Dilatation		
Electrocautery dilation, *n* (%)	10 (83.3)	0
Non-electrocautery dilation, *n* (%)	1 (8.3)	8 (66.7)
Non-dilation, *n* (%)	0	3 (25)
Stent patency, median days	91	101

EUS-CDS: endoscopic ultrasound-guided choledochoduodenostomy, EUS-HGS: endoscopic ultrasound-guided hepaticogastrostomy. SEMS: self-expandable metal stent.

**Table 3 jcm-10-04105-t003:** Adverse events.

	EUS-CDS (*n* = 12)	EUS-HGS (*n* = 12)
Overall adverse events, *n* (%)	1/12 (8.3)	0/12 (0)
Type of adverse events, grade	bile peritonitis, severe	
Early adverse events, *n* (%)	1/12 (8.3)	0/12 (0)
Late adverse events, *n* (%)	0/12 (0)	0/12 (0)

EUS-CDS: endoscopic ultrasound-guided choledochoduodenostomy, EUS-HGS: endoscopic ultrasound-guided hepaticogastrostomy.

**Table 4 jcm-10-04105-t004:** Treatment after the procedures and re-intervention.

	EUS-CDS (*n* = 12)	EUS-HGS (*n* = 12)
Treatment		
Resection, *n* (%)	0	2/12 (16.7)
Chemotherapy	5/12 (41.7)	5/12 (41.7)
Re-intervention, *n* (%)	5 (41.7)	6 (50)
Technical success	5/5 (80)	6/6 (100)
Re-intervention, *n* (%)		
Stent exchange	3/5 (60)	5/6 (83.3)
Stent direction change	2/5 (40)	0
Another EUS-BD	0	1/6 (16.7)

EUS-BD: endoscopic ultrasound-guided biliary drainage. EUS-CDS: endoscopic ultrasound-guided choledochoduodenostomy, EUS-HGS: endoscopic ultrasound-guided hepaticogastrostomy.

## Data Availability

The data presented in this study are available on request from the corresponding author. The data are not publicly available due to ethical and privacy restrictions.

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
