# Peer review of "Outcomes of Endoscopic Ultrasound-Guided Biliary Drainage in a General Hospital for Patients with Endoscopic Retrograde Cholangiopancreatography-Difficult Transpapillary Biliary Drainage"

_jcm, 2021, doi:10.3390/jcm10184105_

Round 1
Reviewer 1 Report
As a retrospective study on the intial EUS-BD experience of a single center, the authors concluded that EUS-BD was possible in general hospitals, not tertiary centers. However, this conclusion is considered to be flawed, and I wonder the conclusion could be generalized. It is necessary to describe the number of ERCP, EUS, and EUS interventions and the learning curve that the author experienced before performing the procedure alone.
Author Response
Reviewer 1
Thank you for a chance to revise our paper.
・As a retrospective study on the intial EUS-BD experience of a single center, the authors concluded that EUS-BD was possible in general hospitals, not tertiary centers. However, this conclusion is considered to be flawed, and I wonder the conclusion could be generalized. It is necessary to describe the number of ERCP, EUS, and EUS interventions and the learning curve that the author experienced before performing the procedure alone.
Thank you for great comment for our paper, author had more than 1000 ERCP experiences and more than 1000 EUS experiences. Author have experienced more than 200 cases of EUS-FNA, but by the time of this study, author had about 10 cases of EUS-BD. Add the above to the manuscript.
Reviewer 2 Report
The authors described the relevancy of EUS-BD for difficult or unsuccessful trans-papillary biliary drainage using ERCP, even in a general hospital. It is well written and an interesting article. However, there are several concerns about revising an article.
(Major)
- The authors described the success rate and safety of EUS-BD in general hospitals in the introduction. The authors described well the success rate of EUS-BD. However, they did not describe the safety of EUS-BD in the main article.
- Would you please explain the definition of the adverse events and more details in the main article (e.g., bleeding, perforation, and so on)?
- The authors should present the table of adverse events separately.
- The authors should discuss the adverse events in the article.
- Page 3, Line 82-87; These sentences should belong to the method. In addition, please define skilled endoscopists (e.g., how many endoscopic procedures)
(Minor)
- Each figure needs explanations of EUS-CDS and EUS-HGS below.
Author Response
Reviewer 2
Thank you for a chance to revise our paper.
The authors described the relevancy of EUS-BD for difficult or unsuccessful trans-papillary biliary drainage using ERCP, even in a general hospital. It is well written and an interesting article. However, there are several concerns about revising an article.
(Major)
1.The authors described the success rate and safety of EUS-BD in general hospitals in the introduction. The authors described well the success rate of EUS-BD. However, they did not describe the safety of EUS-BD in the main article.
Thank you for great comment for our paper, in this study, adverse events were observed in only one case of EUS-CDS, and no adverse events were observed in EUS-HGS. Although it was a small number of studies, the introduction of EUS-HGS is considered safe, so I described it in the discussion.
2. Would you please explain the definition of the adverse events and more details in the main article (e.g., bleeding, perforation, and so on)?
Thank you for great comment for our paper, I added more details to the item of definition of adverse event.
3.The authors should present the table of adverse events separately.
Thank you for great comment for our paper, I changed only to the table of adverse events item.
4.The authors should discuss the adverse events in the article.
Thank you for great comment for our paper, in this study, adverse events were observed in only one case of EUS-CDS, and no adverse events were observed in EUS-HGS. Although it was a small number of studies, the introduction of EUS-HGS is considered safe, so I described it in the discussion.
5.Page 3, Line 82-87; These sentences should belong to the method. In addition, please define skilled endoscopists (e.g., how many endoscopic procedures)
Thank you for great comment for our paper, author had more than 1000 ERCP experiences and more than 1000 EUS experiences. Author has experienced more than 200 cases of EUS-FNA, but by the time of this study, author had about 10 cases of EUS-BD. Add the above to the manuscript in the method.
(Minor)
1.Each figure needs explanations of EUS-CDS and EUS-HGS below.
Thank you for great comment for our paper, I added the explanation of EUS-CDS, EUS-HGS to each figure.
Round 2
Reviewer 2 Report
The article has revied well, it is improved significantly.
Author Response
Reviewer 2
The article has revied well, it is improved significantly.
-Thank you for great comment for our paper.